# Pre-existing bilayer stresses modulate triglyceride accumulation in the ER versus lipid droplets

Valeria Zoni[1†], Rasha Khaddaj[1†], Pablo Campomanes[1], Abdou Rachid Thiam[2], Roger Schneiter[1], Stefano Vanni[1]*

[1]University of Fribourg, Department of Biology, Fribourg, Switzerland; [2]Laboratoire de Physique de l'École Normale Supérieure, ENS, Université PSL, CNRS, Sorbonne Université, Université de Paris, Paris, France

**Abstract** Cells store energy in the form of neutral lipids (NLs) packaged into micrometer-sized organelles named lipid droplets (LDs). These structures emerge from the endoplasmic reticulum (ER) at sites marked by the protein seipin, but the mechanisms regulating their biogenesis remain poorly understood. Using a combination of molecular simulations, yeast genetics, and fluorescence microscopy, we show that interactions between lipids' acyl-chains modulate the propensity of NLs to be stored in LDs, in turn preventing or promoting their accumulation in the ER membrane. Our data suggest that diacylglycerol, which is enriched at sites of LD formation, promotes the packaging of NLs into LDs, together with ER-abundant lipids, such as phosphatidylethanolamine. On the opposite end, short and saturated acyl-chains antagonize fat storage in LDs and promote accumulation of NLs in the ER. Our results provide a new conceptual understanding of LD biogenesis in the context of ER homeostasis and function.

*For correspondence:
stefano.vanni@unifr.ch

[†]These authors contributed equally to this work

Competing interests: The authors declare that no competing interests exist.

## Introduction

Lipid droplets (LDs) are ubiquitous intracellular organelles that consist of a core of neutral lipids (NLs), mostly triglycerides (TG) and sterol esters, surrounded by a phospholipid (PL) monolayer (*Walther and Farese, 2012*). Because of this unique composition, they are the cellular sites responsible for energy and lipid storage and they play a central role in lipid and cellular metabolism (*Walther and Farese, 2012*; *Gao and Goodman, 2015*; *Pol et al., 2014*; *Olzmann and Carvalho, 2019*).

LDs emerge from the endoplasmic reticulum (ER), where the NLs constituting them are synthetized by acyltransferases that are essential for LD formation (*Sandager et al., 2002*). These early LDs have been observed with electron microscopy (EM), showing an oblate lens-like structure with diameters of 40–60 nm (*Choudhary et al., 2015*). Recent experiments suggest that LDs form at specific ER sites marked by the proteins seipin (*Salo et al., 2019*) and Nem1 (*Choudhary et al., 2020*), upon arrival of seipin-interaction partner protein promethin/LDAF1 (LDO in yeast, *Chung et al., 2019*; *Teixeira et al., 2018*; *Bohnert, 2020*; *Eisenberg-Bord et al., 2018*; *Castro et al., 2019*), before a subsequent maturation that involves LD growth and budding (*Pol et al., 2014*; *Olzmann and Carvalho, 2019*; *Walther et al., 2017*; *Joshi et al., 2017*; *Sturley and Hussain, 2012*; *Thiam and Beller, 2017*; *Chen and Goodman, 2017*; *Thiam and Forêt, 2016*).

Knockout of seipin, however, does not abolish LD formation, and, for what concerns the early stages of LD biogenesis, rather results in delayed LD formation and accumulation of NLs in the ER (*Wang et al., 2016*). At the same time, deletion of other proteins with very different functions, such as the lipid phosphatase Pah1 (*Adeyo et al., 2011*) or the membrane-shaping protein Pex30 together with seipin (*Wang et al., 2018*), results in a more pronounced phenotype, with the majority

of cells completely lacking LDs and with NLs strongly accumulating in the ER (*Adeyo et al., 2011*; *Wang et al., 2018*). These observations reveal that LD formation is highly sensitive to the membrane environment, and they indicate that while seipin functions as the main ER nucleation seed for LD biogenesis, other mechanisms are likely to significantly contribute to the energetics of TG accumulation in vivo.

However, because of important challenges in reconstituting in vitro the machinery of LD formation, the early molecular steps of LD biogenesis remain largely unexplored. The mechanistic understanding of these steps (*Pol et al., 2014*; *Olzmann and Carvalho, 2019*; *Walther et al., 2017*; *Joshi et al., 2017*; *Sturley and Hussain, 2012*; *Thiam and Beller, 2017*; *Chen and Goodman, 2017*; *Thiam and Forêt, 2016*; *Santinho et al., 2020*) is currently based on the extrapolation of experimental observations in reconstituted systems, where micrometric oil blisters have been observed, during the formation of black lipid membranes (*White, 1977*; *White, 1986*) or in droplet-embedded vesicle systems (*Ben M'barek et al., 2017*), suggesting that LDs can spontaneously assemble following the accumulation of NLs in between the bilayer leaflets, and that this process can be mediated by membrane properties such as curvature (*Santinho et al., 2020*) or tension (*Ben M'barek et al., 2017*). However, all these experiments often lack the ability to continuously modulate and monitor the NL-to-PL ratio during the process of LD formation, i.e. for increasing values of the NL-to-PL ratio as in cellular conditions, and it is thus difficult to establish the key factors controlling LD assembly based exclusively on information from reconstituted systems. To this end, molecular dynamics (MD) simulations have shown promise as a good alternative strategy to investigate the early stages of LD formation, as this approach has been used to investigate the formation of oil blisters at two different NL-to-PL ratios, showing that NL blister formation takes place when the NL-to-PL ratio exceeds a certain threshold (*Khandelia et al., 2010*), and to investigate how membrane properties alter the propensity of LD egress from the ER bilayer (budding, *Ben M'barek et al., 2017*; *Chorlay et al., 2019*).

Here, we used in silico MD simulations to investigate how membrane lipids modulate the mechanism of fat accumulation. We identified that while certain lipids promote the packaging of NL into oil blisters, others oppose this process. In particular, lipids' acyl chains play a major role in this mechanism. Our results recapitulate several previous experimental observations on LD formation within a unified conceptual framework. Further, in order to validate in vivo our molecular simulations, we took advantage of yeast genetics and we studied fat accumulation in different conditions, where we could rationally modulate this process in agreement with our computational predictions. Our results pave the way for a new conceptual understanding of LD biogenesis in the context of ER homeostasis and function.

## Results

### Diacylglycerol promotes nucleation of TG blisters

In yeast, diacylglycerol (DAG) has been shown to accumulate at sites of LD formation (*Choudhary et al., 2018*). In addition, deletion of Pah1, the lipid phosphatase converting phosphatidic acid (PA) into DAG upstream of TG synthesis, results in a phenotype where the ER shows extensive proliferation and where NLs are present in the cell at significant concentrations but fail to cluster into LD and rather accumulate in the ER (*Adeyo et al., 2011*).

To further characterize the role of DAG on LD biogenesis, we expressed a previously characterized DAG sensor in the ER of yeast cells (*Choudhary et al., 2018*). This sensor is comprised of tandem C1 domains from human protein kinase D (C1a/b-PKD) fused to GFP, which in turn is fused to the transmembrane region of Ubc6, a tail anchored domain protein (*Choudhary et al., 2018*). With this probe, we could compare the localization of DAG in the ER of wild-type cells with that of cells in which NLs fail to be packaged into LDs and are rather spread over the entire ER, such as those where TG is the only NL and Pah1 is deleted. To strictly focus on the PA→DAG→TG pathway, we opted to investigate this behavior in a yeast strain that is devoid of sterol ester. This can be achieved by deleting both acyltransferases responsible for sterol ester formation (*Are1, Are2*) in conjunction with Pah1 (*pah1Δare1Δare2Δ*, *Figure 1A*).

In wild-type cells, the ER-DAG sensor localizes in the ER and is enriched in few puncta per cell (*Figure 1A*). In cells lacking NL synthesis (*dga1Δlro1Δare1Δare2Δ*) the distribution of the ER-DAG

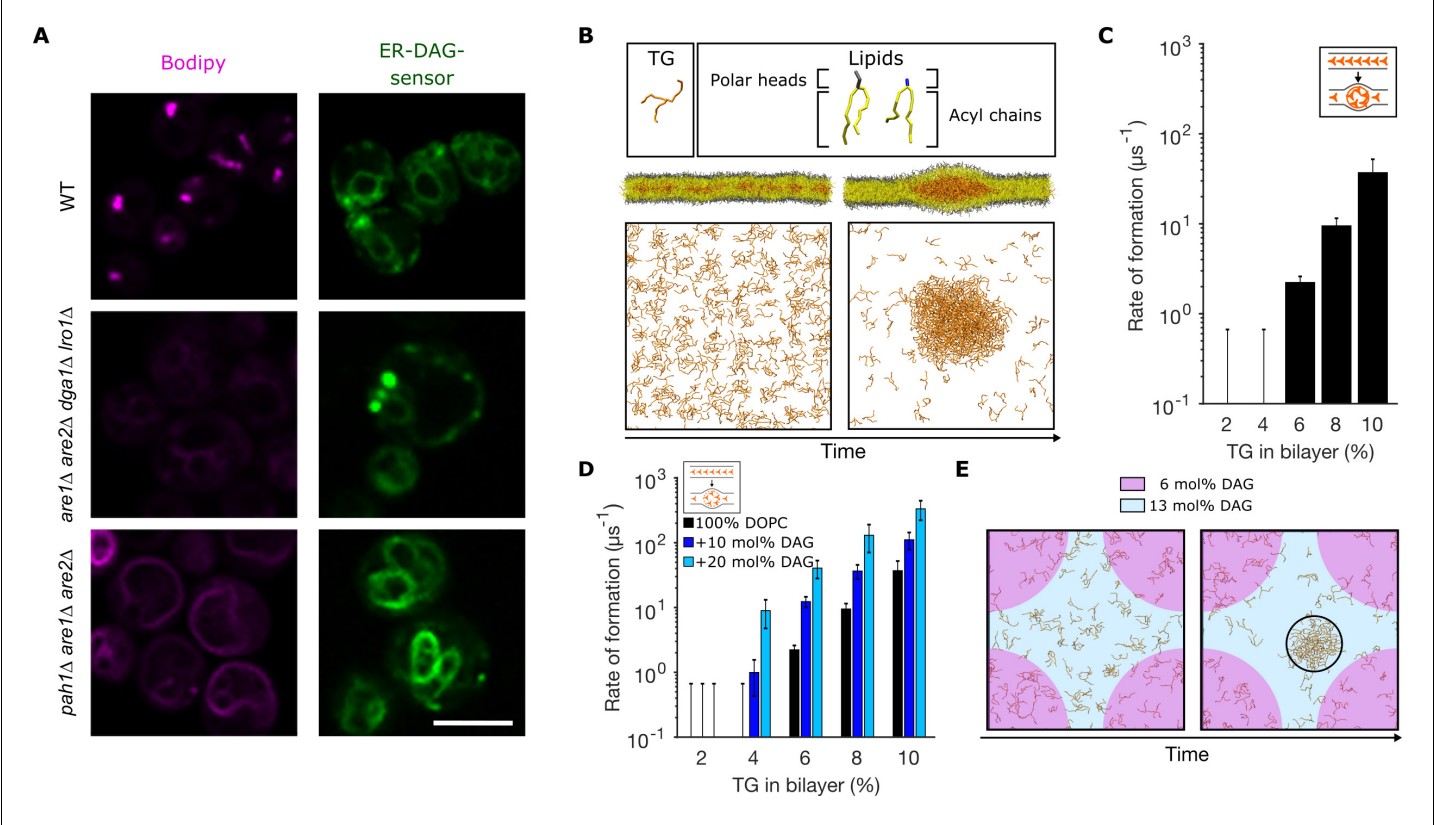

**Figure 1.** Diacylglycerol (DAG) promotes nucleation of triglyceride (TG) blisters. (**A**) Fluorescence microscopy images of WT, *dga1Δlro1Δare1Δare2Δ* *(4Δ)* and *pah1Δare1Δare2Δ* yeast cells. Left panels: neutral lipids (NLs) are stained by BODIPY, right panels: ER-DAG sensor staining. Microscope intensity settings in the three panels are identical to allow for quantitative comparison. (**B**) Setup used to investigate TG nucleation in molecular dynamics (MD) simulations. TG at different concentrations are randomly distributed in a bilayer and simulations are run until formation of blisters or for a total length of 1.5 µs. TG concentrations are reported as ratio between TG and phospholipids (PLs). (**C**) Rate of formation of TG blisters at different TG concentrations, obtained from MD simulations. (**D**) Rate of formation of TG blisters at different TG concentrations in the presence of DAG (10 mol%, 20 mol%), from MD simulations. TG concentrations are reported as ratio between TG and PLs. (**E**) Top view of TG nucleation from simulations of lipid bilayers with two coexisting DAG concentrations (6 mol% and 13 mol%). The two bilayer regions with different DAG concentrations have identical surface areas.

The online version of this article includes the following figure supplement(s) for figure 1:

**Figure supplement 1.** Diacylglycerol (DAG) promotes nucleation of triglyceride (TG) blisters.

sensor is consistent with that of the wild type with few puncta per cells (*Figure 1A*). However, in cells where TG fails to be packaged in LDs and is rather spread in a proliferated ER (as in *pah1Δ* [(*Adeyo et al., 2011*)]), the ER-DAG sensor fails to mark discrete ER sites and rather shows a uniform ER localization (*Figure 1A*).

While the observed defect in LD formation in the *pah1Δ* strain (*Adeyo et al., 2011*) could simply originate from the increase in ER membrane, and hence in larger area for NL diffusion, our data suggest that increased DAG local concentration at specific ER sites might also play a role in LD formation, in agreement with similar results reported while this work was under review, showing that DAG accumulates at LD formation sites marked by seipin and Nem1 (*Choudhary et al., 2020*).

To investigate whether DAG plays a role in promoting LD formation independent of its role in protein recruiting, we opted to reconstitute in silico the process of spontaneous TG blister nucleation previously observed by *Khandelia et al., 2010*. To do so, we built lipid bilayers with TG molecules initially distributed randomly between the two leaflets of the membrane (*Figure 1B*), and we then let the simulations run toward equilibrium until lens formation took place. We observed that, in pure 1,2-dioleoyl-sn-glycero-3-phosphocholine (DOPC) bilayers, lens formation took place starting at TG/PL concentrations above 4% (*Figure 1C*), in agreement with what was previously

reported (*Khandelia et al., 2010*). In addition, we observed that the rate of lens formation is concentration-dependent, as in classical nucleation processes (*Thiam and Forêt, 2016*; *Figure 1C*). Next, we prepared DOPC bilayers with increasing amounts of DAG, and we investigated TG blister formation (*Figure 1D*) in these conditions. Remarkably, we observed a pronounced effect of DAG toward blister formation, with DAG promoting phase separation at low (<4%) TG concentrations and consistently accelerating the rate of blister formation (*Figure 1D*).

Next, to investigate the role of DAG local concentration in this process, we investigated blister formation in a lipid bilayer where, thanks to the use of an external potential (see Materials and methods), half the bilayer has a low (6 mol%) DAG concentration while the other half presents a high (13 mol%) DAG concentration (*Figure 1E* and *Figure 1—figure supplement 1*). In this system, blister formation always occurred in the region with the higher concentration of DAG (*Figure 1E*). These results support the hypothesis that a local increase in the membrane concentration of DAG could promote LD biogenesis in a protein-independent manner, by altering the propensity of TG to condensate into blisters.

## Blister formation is consistent with a phase transition-driven liquid condensation model

Our in silico reconstitution of TG blister formation suggests that this mechanism is consistent with a nucleation process (*Figure 1B and C*). As such, the nucleation energy $E_{nucl}$ is the key parameter that determines its rate, and hence its probability to occur, as the nucleation rate goes as $exp(-E_{nucl}/KT)$ (*Thiam and Forêt, 2016*).

In parallel, from a conceptual point of view, the coexistence of NLs in both the ER membrane and nucleated LDs that is observed in different non-physiological conditions (*Figure 1A* and refs [(*Wang et al., 2016*; *Adeyo et al., 2011*; *Wang et al., 2018*)]) suggests that the process could be described in the framework of a phase separation process, as previously proposed (*Thiam and Forêt, 2016*). In this framework, nucleated fat lenses would constitute the condensed phase, equilibrating with 'free' NLs in the ER membrane.

To investigate whether our in silico approach is consistent with this framework, we next focused on the characterization of the thermodynamic properties of this process. To this end, we sought to determine the equilibrium concentration of TG molecules in the bilayer (i.e. their chemical potential) when the bilayer is in equilibrium with a TG droplet large enough to mimic the properties of a bonafide condensed phase.

To do so, we first prepared a system that allowed us to estimate the concentration of TG molecules in a lipid bilayer by packing all TG molecules in a pre-formed lens inside a lipid bilayer, and we followed the time evolution of the concentration of TG molecules spreading into the bilayer (*Figure 2A and B* and *Figure 2—figure supplement 1*). After an initial equilibration, the system reached equilibrium at the concentration of 1.1 ± 0.1% (*Figure 2B*), in close agreement with experimental values reported using capacitance measurements (*White, 1986*) or nuclear magnetic resonance (NMR) (*Hamilton et al., 1983*).

To investigate whether the process is consistent with equilibrium phase coexistence, we used three different approaches. First, we arbitrarily increased the concentration of the diluted TG in the bilayer by computationally 'injecting' new TG molecules in an equilibrated system (*Figure 2C*). When the system reached equilibrium, all excess TG molecules translocated to the oil lens, with the bilayer TG concentration returning to its initial value after few microseconds (*Figure 2C and D*).

Second, to mimic LD growth and to validate whether the TG blister is representative of the condensed phase, we prepared oil lenses of different sizes compatible with the ones observed using EM (*Choudhary et al., 2015*), ranging from 25 to 50 nm in diameter (*Figure 2E*). We observed that the concentration of equilibrium TG in the bilayer is independent of the blister size, further suggesting that our pre-existing TG droplet is large enough to mimic the properties of a bona-fide condensed phase.

Third, when TG blisters with a total TG concentration below the computed chemical potential (1.1% TG/PL) were simulated, this led to blister dissolution (*Figure 2F* and *Video 1*). Taken together, our results indicate that, in our MD simulations, TG blister formation in lipid bilayers is consistent with equilibrium phase coexistence between the blister and the bilayer.

We next used this approach to estimate the effect of DAG in modulating the equilibrium concentration of TG in bilayers. We observed that the presence of DAG depletes TG from lipid bilayers

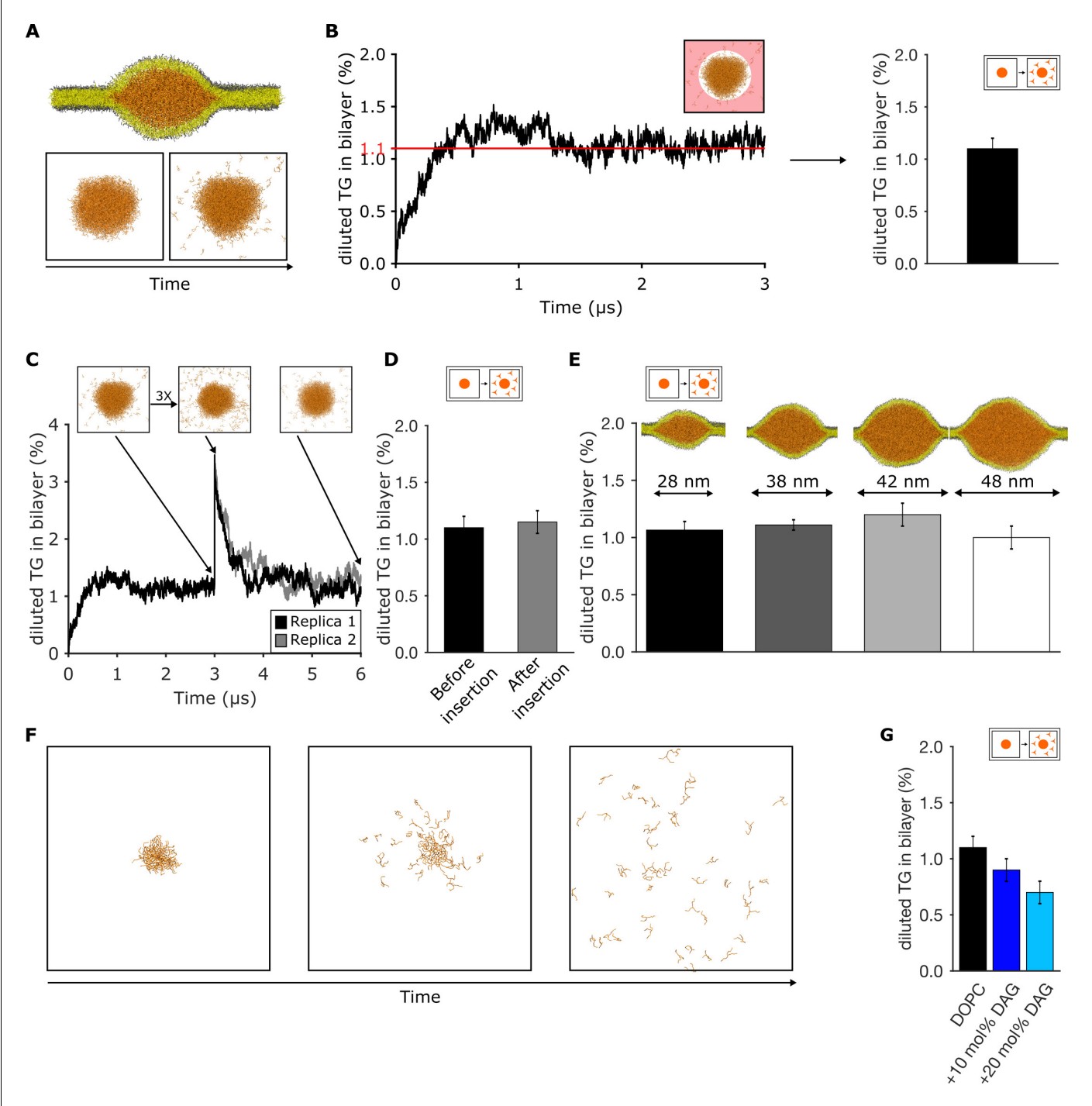

**Figure 2.** Triglyceride (TG) blister formation in lipid bilayers is consistent with a phase separation process. (**A**) Setup used to measure the amount of diluted TG via spontaneous diffusion from a pre-formed blister and (**B**) quantification over time. The area in which TG is assumed as diluted is highlighted in light red. (**C**) Time evolution of the percentage of diluted TG inside the bilayer. The injection of additional TG molecules was performed after 3 μs of dynamics. (**D**) Comparison between the values of diluted TG before and after the insertion of TG in the bilayer. (**E**) Quantification of diluted TG in oil blisters of different sizes. (**F**) Time evolution of blister dissolution when the total TG concentration in the system is below the threshold observed in (**B and C**). (**G**) Percentage of diluted TG in lipid bilayers enriched in diacylglycerol (DAG) lipids. In all the panels, TG concentrations are reported as ratio between TG and phospholipid (PL).

The online version of this article includes the following figure supplement(s) for figure 2:

**Figure supplement 1.** Choice of the radius for the calculation of diluted triglycerides (TG).

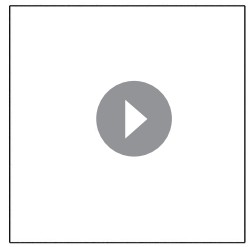

**Video 1.** Triglyceride (TG) lens dissolution in a DOPC bilayer. The concentration of TG is below (<1%) the calculated 'free TG' threshold in a DOPC bilayer (1.1 ± 0.1%).
https://elifesciences.org/articles/62886#video1

(*Figure 2G*). This result parallels our previous observation that DAG promotes blister formation (*Figure 1D*), and it provides a mechanistic explanation of why deletion of Pah1, the lipid phosphatase responsible for the synthesis of DAG, results in NL accumulation in a proliferated ER (*Figure 1* and ref [(*Adeyo et al., 2011*)]).

## The ER PL content is optimal in decreasing free TG levels and promoting TG blister formation

The observation that DAG concomitantly promotes TG blister formation and depletion of TG molecules from the ER is intriguing. To further characterize the relationship between blister formation and TG chemical potential, we next focused on the role of membrane lipids on these processes, with the aim of potentially establishing a correspondence with the role of ER membrane in modulating LD formation.

First, we investigated the role of phosphatidyl-ethanolamine (PE), a PL that is specifically enriched in the ER membrane, including at LDs (*Figure 3A*; *Penno et al., 2013*; *Bartz et al., 2007*). We observed that PE promotes blister formation (*Figure 3B*) and, correspondingly, depletion of TG from the membrane (*Figure 3C*). Interestingly, this is quite significant already at physiological-like concentration of PE lipids (≈30%, *van Meer et al., 2008*).

Next, we investigated the role of acyl chain saturation, as PLs carrying unsaturated chains have been shown to be particularly enriched in the ER membrane (*van Meer et al., 2008*; *Harayama and Riezman, 2018*). By preparing lipid bilayers with PLs with increasing levels of saturated acyl-chains (*Figure 3D*), we observed that lipid saturation slightly opposes blister formation (*Figure 3E*) and, correspondingly, promotes accumulation of TG in the membrane (*Figure 3F*).

We next investigated blister formation and TG accumulation in the presence of lipids that are scarce in the ER and that have a very different chemistry, including short-chain lipids (*Figure 3G*) and cholesterol (*Figure 3J*). We found that regardless of lipid polar head, shorter chains favor the accumulation of TG molecules in the bilayer (*Figure 3I*) and prevent blister formation (*Figure 3H*), while cholesterol, on the other hand, promotes blister formation (*Figure 3K*) and depletes TG from the bilayer (*Figure 3L*), in agreement with previously reported values using NMR (*Spooner and Small, 1987*).

Lastly, we prepared bilayers with a lipid mixture that approximates that of the ER (*Figure 3M*), by including PL unsaturation, high PE content, and low but non-negligible quantities of DAG and cholesterol (*van Meer et al., 2008*). As it can be seen in *Figure 3N and O*, the effect of the various lipids adds up, resulting in a dramatic acceleration of blister formation. Taken together, our results suggest that the ER membrane appears engineered to promote oil blister formation and to allow only very low concentration of diluted TG in the ER, possibly to reduce TG-induced lipotoxicity.

## TG chemical potential is a good descriptor of blister formation kinetics

Our data suggest that bilayer properties modulate TG blister formation in silico, and possibly in vivo, consistently with equilibrium phase coexistence between the blister and the bilayer. To this end, our simulations are consistent with theoretical models based on continuum approaches (*Thiam and Forêt, 2016*; *Zanghellini et al., 2010*) suggesting that LD biogenesis might take place upon spontaneous de-mixing of TG molecules in PL bilayers. In addition, however, our approach allows one to test and quantify some of the predictions put forward by these models, including whether PL de-mixing alone is sufficient to promote LD biogenesis (*Zanghellini et al., 2010*) and to quantify to what extent the interaction energy between TG and PLs (*Thiam and Forêt, 2016*) modulates LD formation.

In our simulations of nascent TG blisters in the presence of multicomponent PL composition, we indeed observed spontaneous de-mixing of PLs according to their intrinsic curvature, with PE and

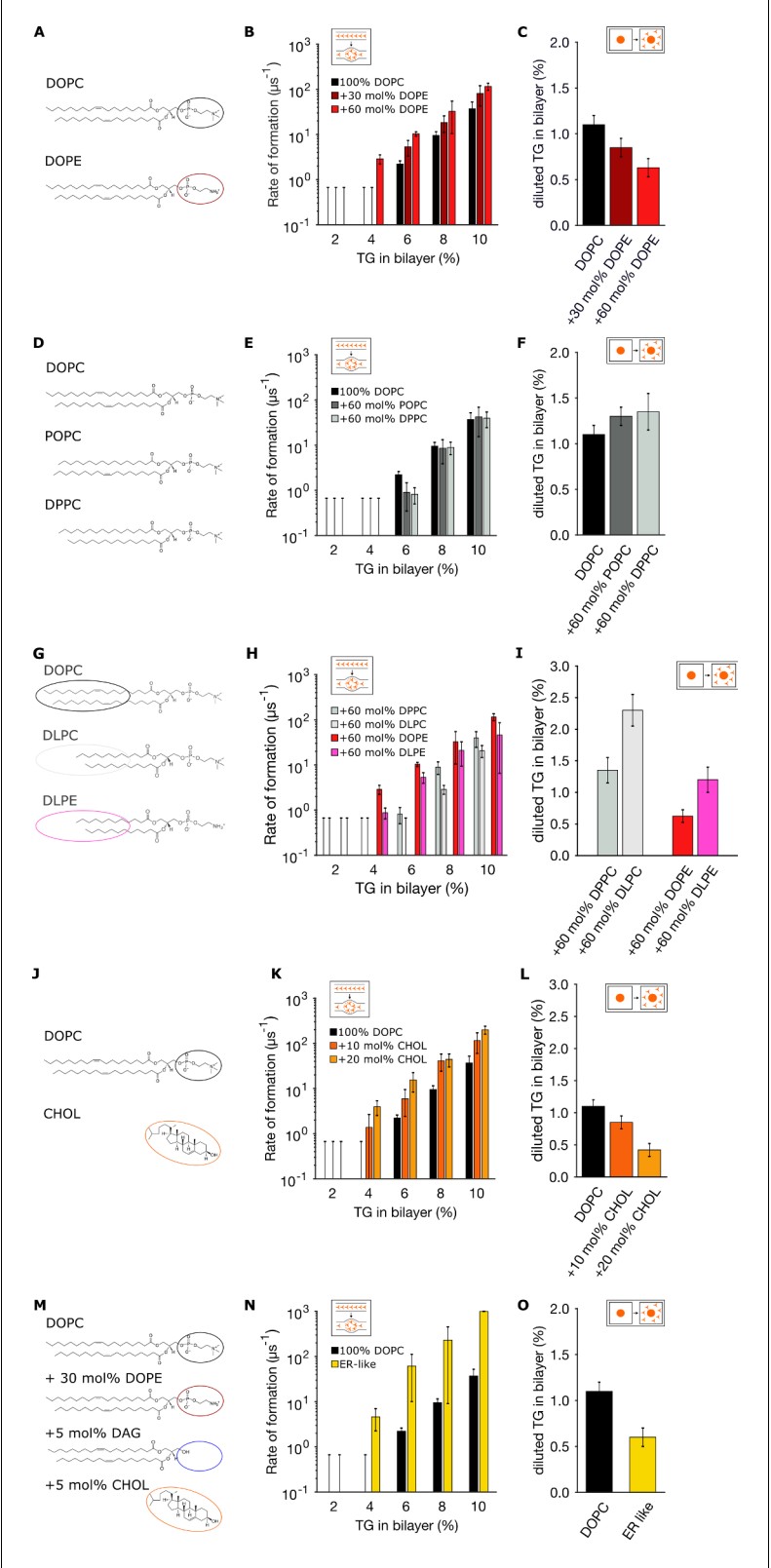

**Figure 3.** Endoplasmic reticulum (ER) membrane lipids promote triglyceride (TG) blister formation. (A, D, G, J, M) Lipid compositions tested and chemical structures of the various lipids involved in the mixtures. (B, E, H, K, N) Corresponding rates of blister formation, and (C, F, I, L, O) percentage of diluted TG. In all the panels, TG concentrations are reported as ratio between TG and phospholipid (PL).

DAG accumulating in the region of negative curvature surrounding the TG blister (*Figure 4A and B*). At the same time, however, also PL mixtures consisting of saturated (1,2-dipalmitoyl-sn-glycero-3-phosphocholine, DPPC) and unsaturated (DOPC) lipids show PL de-mixing (*Figure 4C*) even if in those conditions blister formation is opposed with respect to pure DOPC lipid bilayers. Thus, our

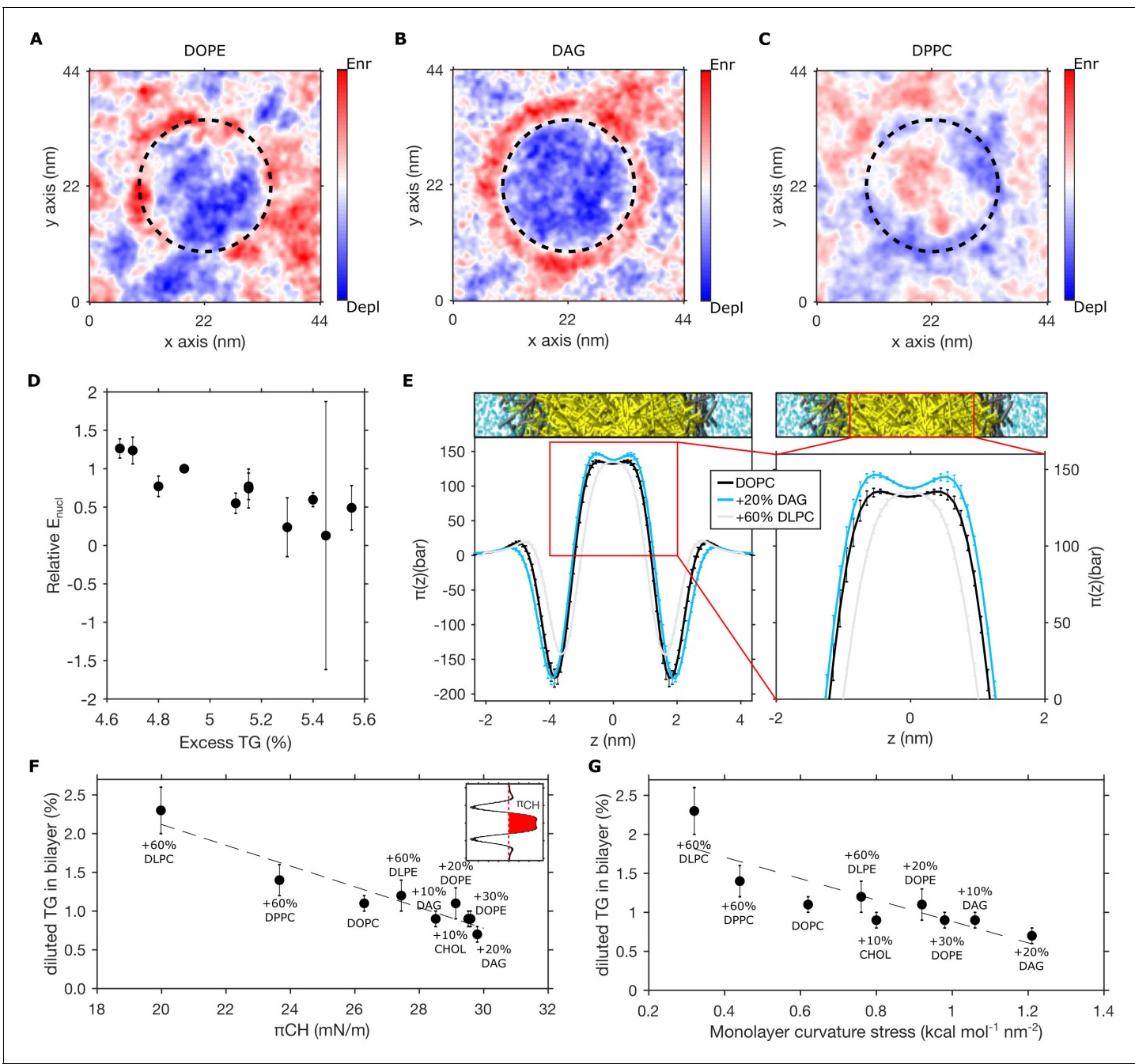

**Figure 4.** Pre-existing bilayer stresses modulate in-bilayer diluted triglyceride (TG) concentration. (A–C) Depletion-enrichment maps for (A) DOPC + 20 mol% DOPE, (B) DOPC + 20 mol% diacylglycerol (DAG), and (C) DOPC + 60 mol% DPPC bilayers in the presence of a TG blister (dashed line). (D) Correlation between the relative nucleation energy ($E_{nucl}$) for different bilayer compositions (with respect to $E_{nucl}$ of DOPC bilayers) and excess TG. (E) Lateral pressure profile of bilayers containing different lipid compositions. (F and G) Correlation between chain pressure $\pi_{CH}$ (F) or monolayer curvature stress (G) and TG equilibrium concentration for bilayer mixtures containing DOPC, DOPE, DAG, DPPC, DLPC, DLPE, and cholesterol. All the percentage in the graph referring to bilayer compositions are to be intended as mol%, while TG concentrations are reported as ratio between TG and phospholipid (PL).

simulations suggest that PL de-mixing per se (*Zanghellini et al., 2010*) is not sufficient to promote TG blister formation.

On the other hand, the nucleation energy for nascent TG blisters can be described as a sum of two linked contributions, $E_{nucl} = E_{\Delta\mu} + E_s$, with $E_{\Delta\mu}$ arising from TG de-mixing and $E_s$ arising from interfacial and mechanical contributions (*Thiam and Forêt, 2016*). Hence, the effect of PLs on phase separation can be both on TG chemical potential and membrane mechanics: PLs may induce curvature stress in the bilayer that could potentially alter TG chemical potential and generate interactions with TG that could alter both monolayer and bilayer bending rigidity.

We estimated the nucleation energy $E_{nucl}$ from our MD simulations of spontaneous blister formation and we plotted it, for the various lipid compositions tested in this study, as a function of the excess of TG, i.e. the difference between the TG% used in the nucleation setup (6%) and the corresponding TG chemical potential (*Figure 4D*). The nucleation energy decreased with excess TG in the bilayer, which agrees with the description of nucleation phenomena. Therefore, this observation supports that the simple increase in TG bilayer chemical potential, here, through increasing TG bilayer concentration, is sufficient to stimulate LD nucleation. Hence, the observed changes in nucleation rate as a function of lipid composition pertain, at least partly, to a variation in TG bilayer chemical potential (*Figure 4D*). This model agrees with recent experimental results (*Santinho et al., 2020*) showing that the effect of membrane curvature in LD formation originates from its ability to alter the critical TG concentration required for the spontaneous condensation of TG.

## TG chemical potential is modulated by pre-existing bilayer stresses

Our above model suggests that both thermodynamic and kinetic properties of TG-in-bilayer systems can be interpreted by a change in TG chemical potential in the diluted phase. Such changes may be induced by specific TG–PL interactions or pre-existing bilayer stresses. Therefore, we investigated how these two factors modulate the equilibrium concentration of TG in bilayers.

To do so, we next computed the lateral pressure profile (LPP) of the same lipid bilayers previously described, but in the absence of TG molecules, with the aim of identifying molecular properties that would correlate with their equilibrium TG concentration. In short, the LPP describes the distribution of lateral stresses across a PL bilayer that arise from the interactions between the lipid molecules, and it provides a direct connection between molecular (e.g. lipid intrinsic curvature and lipid–lipid interactions) and large-scale (e.g. surface tension and bending rigidity) properties of PL (*Marsh, 2007*).

We observed that lipids showing a marked depletion of TG from the membrane, such as DAG, induce a substantial increase in the positive pressure between the hydrophobic chains with respect to pure DOPC bilayers (*Figure 4E*). Of note, while LPPs from CG simulations do not have the same level of detail of atomistic ones, this increase in lipid chain pressure by DAG (and PE) is consistent with previously reported LPPs using atomistic simulations (*Vamparys et al., 2013*; *Ding et al., 2015*). On the other hand, the bilayer conditions we found to oppose TG blister formation, such as the presence of short-chain lipids, display the opposite behavior, i.e. a decrease in lipid chain pressure (*Figure 4E*).

To better quantify the relationship between TG concentration and lipids' acyl chains properties, we plotted the equilibrium TG concentration computed in our simulations against two distinct properties: (i) the hydrophobic pressure $\pi_{CH}$, i.e. the integral of the LPP corresponding to the area of the positive region ($\pi(z) > 0$) between both glycerol minima in the LPP (*Figure 4F*; *Marsh, 1996*) and (ii) the total monolayer curvature stress $\kappa_b\,c_0^2$, where $\kappa_b$ is the monolayer bending rigidity and $c_0$ its spontaneous curvature (*Figure 4G*).

In both cases, we found a very good correlation, suggesting that pre-existing stresses in the bilayer can predict the propensity of the bilayer to accept increasing concentration of TG molecules, and hence oppose or promote their spontaneous de-mixing to form oil blisters. In addition, the observation that TG chemical potential and monolayer curvature stress are correlated (*Figure 4G*) confirms that mechanical and de-mixing energetic contributions are not independent (*Thiam and Forêt, 2016*), as TG de-mixing is modulated by bilayer mechanical properties.

Overall, the correlations we observed between TG chemical potential and bilayer stresses provide some semi-quantitative interpretations, with curvature stress allowing to predict the effect on TG accumulation based on PL intrinsic curvature in the absence of explicit LPP calculations, and with the correlation with hydrophobic pressure suggesting that changes in bilayer pressure originating from

the repulsion among PL hydrophobic chains are proportional to changes in chemical potential, via a Gibbs–Duhem relationship.

## Acyl-chain saturation promotes TG accumulation in the ER

Recent experimental and computational results suggest that nucleation is largely driven by protein activity in vivo (*Zoni et al., 2020 Prasanna et al., 2021*). On the other hand, it remains unclear whether variations in PL content could alter TG concentration in the ER. Thus, to test semi-quantitatively our hypothesis that PL modulate TG-in bilayer chemical potential in vivo, we resorted to fluorescence microscopy experiments in yeast, a model system that has been widely used to investigate LD biogenesis. We reasoned that if we could modulate the balance of NLs between LDs and the ER membrane (i.e. between the condensed and diluted phases) with independent approaches consistently with our in silico predictions, this would provide additional evidence in support of the quality of our modeling approach.

To do so, we first investigated the equilibrium between NLs in the ER *vs* LDs in yeast cells with an altered ratio between saturated and unsaturated fatty acids, as our simulations suggest (*Figure 3D–F*) that saturated PLs slightly decrease LD formation and promote accumulation of TG in the lipid bilayer. Of note, unlike DAG lipids that accumulate at sites of LD biogenesis due to local enrichment of DAG-generating enzymes (*Choudhary et al., 2020*), lipids carrying unsaturated chains are unlikely to accumulate locally at sites of LD formation, but should rather spread somewhat homogeneously in the ER membrane.

Since fatty acids are incorporated in both PLs and TG during lipid remodeling, we first investigated in silico the effect of increasing acyl chain saturation in TG (*Figure 5*). Our results suggest that, like for PLs, increasing TG saturation results in TG accumulation in the bilayer (*Figure 5B*). Notably, increasing acyl chain saturation in TG molecules has a larger effect on the amount of diluted TG in comparison with increasing acyl chain saturation in PLs.

Next, we used a temperature-sensitive allele of *Ole1*, a gene that is essential for production of monounsaturated fatty acids as it expresses the lone delta-9 desaturase enzyme in yeast (*Stukey et al., 1989*). This temperature-sensitive allele results in intact proteins with impaired desaturase activity already at permissive temperature (24°C), with increased C16:0 and decreased C18:1 levels (*Tatzer et al., 2002*) and inactive proteins at non-permissive temperature (37°C) with a major decrease in the ratio between unsaturated and saturated fatty acids in the cell (*Stewart and Yaffe, 1991*).

Using fluorescence microscopy, we observed BODIPY-positive ER structures at the non-permissive temperature, indicating that the amount of NLs present in the ER increases as a result of the increase in saturated cellular fatty acids in comparison to wild type (*Figure 5C and D*). This enrichment was promoted by supplementing the cells with the saturated palmitic acid (*Figure 5C and D*), while it was entirely rescued by supplementing the cells with the unsaturated palmitoleic acid (*Figure 5C and D*). To further quantify this behavior, we opted to measure the fluorescent intensities of LDs and membrane (normalized to the total cell intensity) in the different conditions, and we could confirm that an increase in the ratio between saturated and unsaturated fatty acids leads to an increased accumulation of NLs in the membrane, in agreement with our in silico data (*Figure 5E and F*).

## Short-chain lipids promote TG accumulation in the ER

Next, we investigated whether short-chain lipids could alter the equilibrium between LD formation and NL accumulation in the ER in yeast, as we found that, in silico, those lipids result in significant accumulation of TG in the bilayer (*Figure 3*).

To test whether the accumulation of short-chain containing lipids affects NL distribution in vivo, we used a yeast mutant defective in acyl chain elongation (*Elo1Δ*). *Elo1Δ* mutant cells were cultivated in media supplemented either with lauric acid (C12:0) or palmitic acid (C16:0) and NL distribution between the ER and LDs was analyzed by staining with BODIPY. *Elo1Δ* mutant cells displayed increased membrane-associated BODIPY fluorescence when cultivated in the presence of lauric acid compared to palmitic acid, suggesting that NL accumulate in the ER in cells containing short-chain PLs (*Figure 6*).

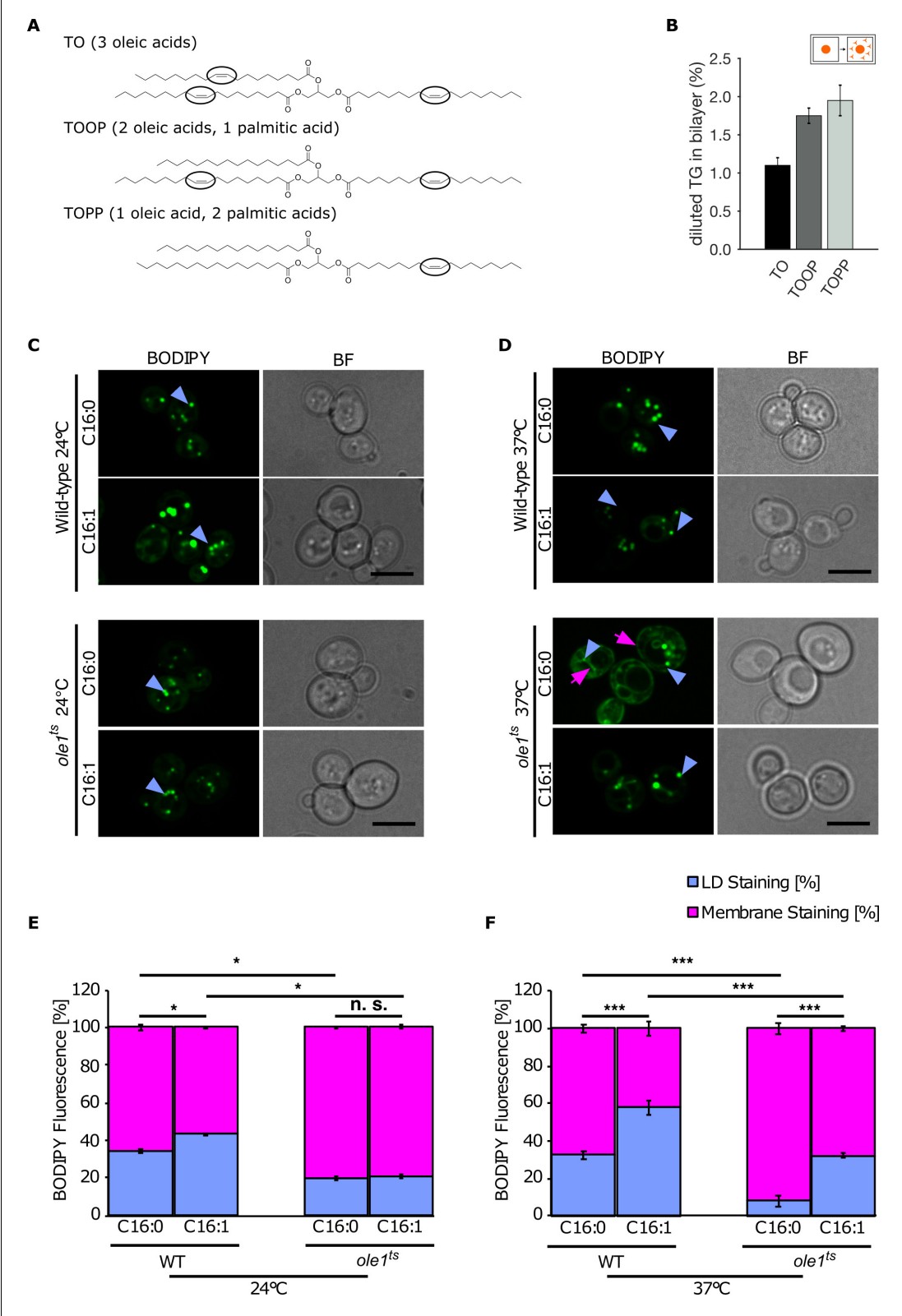

**Figure 5.** Lipid saturation promotes triglyceride (TG) accumulation in the endoplasmic reticulum (ER). (**A**) Chemical structure of unsaturated TG used in molecular dynamics (MD) simulations. Black ovals indicate the presence of double bonds in the acyl chain. (**B**) Equilibrium 'diluted' concentration of different TG in DOPC bilayers. TG concentrations are reported as ratio between TG and phospholipids (PLs). (**C and D**) Fluorescence microscopy images of wild-type and temperature-sensitive *ole1*[ts] at 24°C (**C**) and 37°C (**D**). Strains were cultivated in SC medium containing palmitic acid (C16:0) or

*Figure 5 continued on next page*

Figure 5 continued

palmitoleic acid (C16:1) and cells were stained with the neutral lipid (NL) marker BODIPY. The blue arrowheads highlight lipid droplets (LDs), and the pink arrows point to the cortical and perinuclear ER membrane. (E and F) Quantification of fluorescent intensities of LDs and membranes (n = 50) in the different conditions shown in (C and D): Values represent fluorescent intensity relative to total cellular fluorescence. Asterisks denote statistical significance (Student's t-test, *p < 0.05, ***p < 0.001), n.s., non-significant.

In summary, the agreement between the experimental results and our computational predictions indicates that our modeling approach, despite its inherent simplicity, is able to faithfully reproduce some of the underlying molecular properties of LD formation in vivo, and it points to an important role of ER lipid composition, and especially of its internal stresses, in this biological process.

## Discussion

The current model of LD formation posits that after synthesis in the ER membrane, TG molecules accumulates at ER sites marked by seipin (*Salo et al., 2019*; *Choudhary et al., 2020*; *Chung et al., 2019*), that then acts as a nucleation seed for the formation of a nascent TG lens that will subsequently grow into a mature LD. Alterations in lipid homeostasis result in defects in LD formation (*Adeyo et al., 2011*; *Santinho et al., 2020*; *Cohen et al., 2015*; *Vevea et al., 2015*; *Wolinski et al., 2015*) in some cases promoting non-negligible accumulation of TG in the ER (*Adeyo et al., 2011*; *Cartwright et al., 2015*) and alterations in ER morphology (*Adeyo et al., 2011*; *Vevea et al., 2015*; *Wolinski et al., 2015*). On one hand, these experiments suggest that a detectable accumulation of TG in the ER is a hallmark of non-physiological conditions, under which proper LD formation does not take place; on the other hand, they indicate that seipin-independent ER properties, such as its lipid composition, can significantly alter the propensity of LD formation.

Our data provide a molecular explanation for these experimental observations, and they are consistent with previously proposed theoretical considerations suggesting that the spontaneous de-mixing of TG in the ER can lead to LD formation consistently with a phase separation process (*Thiam and Forêt, 2016*). Thus, while increasing attention is being directed at the role of the protein machinery that localizes at the site of LD formation (*Salo et al., 2019*; *Choudhary et al., 2020*; *Chung et al., 2019*; *Wang et al., 2018*), our results provide further evidence in support of a mechanism of LD formation where proteins act as the main nucleation factor while PLs modulate the propensity for NL de-mixing from the ER membrane.

Importantly, our data suggest that numerous experimental observations can be interpreted in molecular terms by considering the equilibrium concentration of TG in the ER membrane as a major factor in determining the propensity of LD formation. As a consequence, while later stages of LD biogenesis, such as LD budding, can be satisfactorily described using continuum theory approaches (*Ben M'barek et al., 2017*; *Choudhary et al., 2018*), the early stages of LD formation appear to be dominated by the specific molecular interactions between PLs and TG, and thus appear better suited to be investigated with chemical specific particle-based approaches such as MD simulations.

Our observations have important implications for both ER homeostasis and LD biogenesis and growth. First, our data suggest that the amount of TG in the ER at equilibrium is not zero. This has potential implications in the catabolism and homeostasis of TG as it opens the possibility that lipases or lipid transport proteins might interact with TG directly from the ER, rather than upon binding to LD. Second, our model suggests that when a nascent LD is formed, all TG in excess of its equilibrium chemical potential in the ER will flow to existing LD without the need for additional external energy, as long as the protein machinery around the LD-ER contact site does not prevent free diffusion of TG. This is particularly important as pre-existing LDs that are resistant to starvation have been observed, for example, in COS-1 cells (*Kassan et al., 2013*).

Our data suggest that pre-existing lateral stresses in the hydrophobic core of the membrane accurately predict the propensity of TG to condensate in LDs. Remarkably, the specific composition of the ER membrane, and namely high levels of both unsaturated lipids and phosphatidylethanolamine (*van Meer et al., 2008*; *Upadhyaya and Sheetz, 2004*), contribute to keep TG in the ER at very low concentrations, thus not only promoting their packaging into LD, but also potentially preventing ER stress and lipotoxicity (*Listenberger et al., 2003*; *Ertunc and*

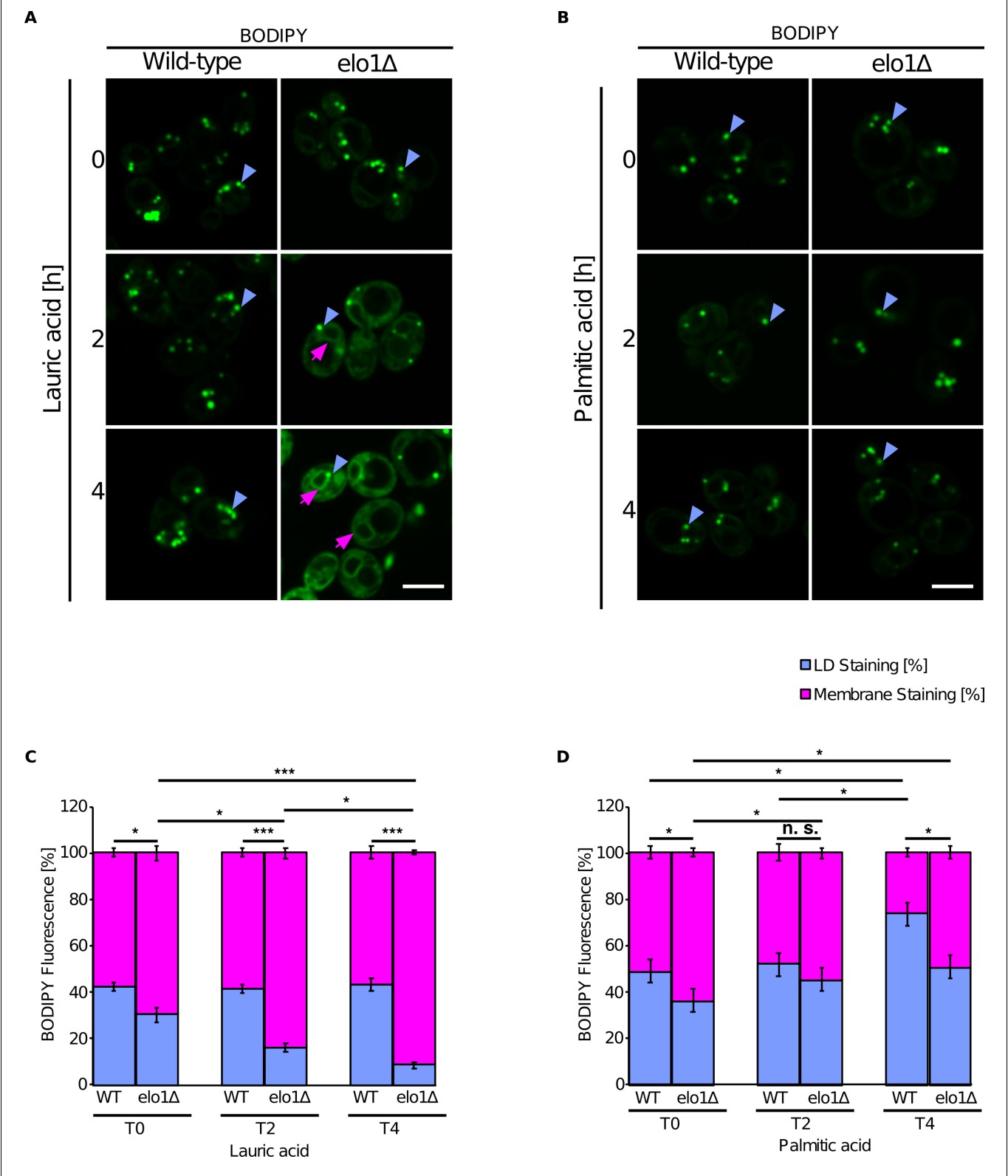

**Figure 6.** Short-chain lipids promote triglyceride (TG) accumulation in the endoplasmic reticulum (ER). (**A and B**) Fluorescence microscopy images of wild-type and *elo1Δ* mutant cells cultivated in SC medium containing lauric acid (C12:0, 2 mM) or palmitic acid (C16:0, 2 mM) for the indicated period of time. Cells were stained with BODIPY and visualized by fluorescence microscopy. Lipid droplets (LDs) are indicated by blue arrowheads and ER stained with BODIPY is marked by purple arrows. Scale bar, 5 µM. (**C and D**) Quantification of BODIPY fluorescence of LD and cellular membranes (n = 50).

*Figure 6 continued on next page*

*Figure 6 continued*

Values represent fluorescent intensity relative to total cellular fluorescence. Asterisks denote statistical significance (Student's t-test, *p < 0.05, ***p < 0.001), n.s., non-significant.

*Hotamisligil, 2016*). Interestingly, we observed that an increase in the amount of saturated fatty acids in yeast cells promotes TG accumulation in the ER. This could potentially provide a new molecular explanation of the elevated toxicity of saturated fatty acids that is independent to the efficiency of their incorporation into TG (*Listenberger et al., 2003*; *Choi et al., 2011*; *Leamy et al., 2016*). In addition, our model could help explain how fat accumulation could be modulated in response to environmental variations, with cells potentially adjusting the local composition of the ER, for example, to promote or delay LD formation in response to the intake of different nutrients.

Our model suggests that an elevated local concentration of DAG, the natural TG precursor, can promote LD formation by virtue of unfavorable interactions arising between DAG and TG when both are present in a lipid membrane. Thus, upon PA to DAG conversion, not only TG synthesis increases, as more substrate becomes available, but also the physical prowess of the ER membrane for TG packaging increases, leading to a positive feedback loop. Quite notably, in order to proceed to later stages of LD formation, and namely LD budding from the ER, the concentration of DAG has to decrease (*Choudhary et al., 2018*). Thus, our data suggest that DAG can be considered as a master regulator of the various stages of LD biogenesis.

A potential limitation of our approach is the description of molecular systems using a CG force field that inherently introduces quantitative inaccuracies in the reproduction of molecular interactions. In this work, this is particularly relevant for what concerns the ability of our model to correctly reproduce the LPP, and the corresponding bilayer stresses, for the various lipids investigated in this study. However, the good agreement between our CG LPP and atomistic LPP available in the literature (*Vamparys et al., 2013*; *Ding et al., 2015*) is very encouraging for the overall quality of our modeling approach. In addition, our experimental strategy does not allow us to investigate directly in situ the mechanism of LD formation, but rather focuses on the quantification of TG accumulation in the ER membrane when cells are starved for essential fatty acids, i.e. *ole1^{ts}* at 37°C for unsaturated lipids and *elo1Δ* for long acyl chains when grown in C12 medium. As such, we cannot quantify to what extent the observed reduction in LD number is caused by a defect in LD formation or by enhanced induction of lipolysis as a response to starvation for essential fatty acids. However, our observation that TG blister formation in silico, like in vitro (*Thiam and Forêt, 2016*), takes place following a phase separation mechanism, strongly indicates that TG accumulation in the ER is very likely coupled with a defect in LD formation, and *vice versa*.

In summary, our results provide a new conceptual view on the mechanism of LD biogenesis and fat accumulation, whereby not only local events taking place at the specific sites of LD formation, but also more global properties of the surrounding ER membrane play a major role in the overall regulation of this process. On one hand, these properties are easier to measure, for example by means of genetic manipulation of lipid composition coupled with lipid imaging techniques, than local properties such as LD surface or line tension. On the other hand, our framework suggests that physiological variations in ER lipid composition at sites spatially distinct from LDs, for example by means of lipid transport or establishment of contact sites with other organelles, might ultimately provide an alternative cellular route to modulate intracellular fat accumulation.

## Materials and methods

**Key resources table**

| Reagent type (species) or resource | Designation | Source or reference | Identifiers | Additional information |
|---|---|---|---|---|
| Strain, strain background (*S. cerevisiae*) | BY4741 | Euroscarf | RRID: SCR_003093 | *Mata, his3Δ1, leu2Δ0, met15Δ0, ura3Δ0.* |

*Continued on next page*

*Continued*

| Reagent type (species) or resource | Designation | Source or reference | Identifiers | Additional information |
|---|---|---|---|---|
| Chemical compound, drug | Palmitic acid | Sigma-Aldrich | P0500 RRID:SCR_008988 | 2 mM |
| Chemical compound, drug | Palmitoleic acid | Sigma-Aldrich | P9417 RRID:SCR_008988 | 2 mM |
| Chemical compound, drug | Lauric acid | Sigma-Aldrich | 8053330100 RRID:SCR_008988 | 2 mM |
| Chemical compound, drug | Brij58 | Sigma-Aldrich | P5884 RRID:SCR_008988 | 1% |
| Software, algorithm | VisiView | Visitron systems GMBH | | 4.2.0 |
| Software, algorithm | ImageJ | https://imagej.nih.gov/ij/ | RRID:SCR_003070 | *Schneider et al., 2012* |
| Software, algorithm | Photoshop | Adobe, Mountain View | RRID:SCR_014199 | |
| Software, algorithm | LAMMPS | https://lammps.sandia.gov | RRID:SCR_015240 | MD simulations software |
| Software, algorithm | GROMACS | http://www.gromacs.org | RRID:SCR_014565 | MD simulations software |
| Software, algorithm | VMD | http://www.ks.uiuc.edu/Research/vmd/ | RRID:SCR_001820 | |
| Other | BODIPY493/503 stain | Invitrogen | D3922 RRID:SCR_008988 | 1 µg/mL |

## Molecular dynamics (MD) simulations

All MD simulations were performed using the software LAMMPS (*Plimpton, 1995*) and employing the Shinoda–Devane–Klein (SDK) force field (*Shinoda et al., 2007*; *Bacle et al., 2017*; *MacDermaid et al., 2015*; *Campomanes et al., 2019*). Parameters for TG were taken from (*Bacle et al., 2017*) and triolein was chosen as the model TG, unless explicitly otherwise stated (*Figure 5A and B*), while 1-2-dioleoyl-sn-glycerol has been used as model for DAG. Non-bonded interactions of DAG with 1,2-dioleoyl-sn-glycero-3-phosphoethanolamine (DOPE) and cholesterol were obtained by combination rules, as described in *MacDermaid et al., 2015* and are listed in *Supplementary file 1*. Initial configurations and input files were obtained through conversion of atomistic snapshots using the CG-it software (https://github.com/CG-it/).

In the simulations, temperature and pressure were controlled via a Nosé–Hoover thermostat (*Nosé, 1984*) and barostat (*Parrinello and Rahman, 1981*; *Martyna et al., 1994*; *Shinoda et al., 2004*): target temperature was 310 K and average pressure was 1 atm. The lateral $xy$ dimensions were coupled, while the $z$ dimension was allowed to fluctuate independently. Temperature was dumped every 0.4 ps, while pressure every 1 ps. Linear and angular momenta were removed every 100 timesteps. Van der Waals and electrostatic interactions were truncated at 1.5 nm. Long-range electrostatics beyond this cutoff were computed using the particle-particle-particle-mesh (PPPM) solver, with an RMS force error of $10^{-5}$ kcal mol$^{-1}$ Å$^{-1}$ and order 3. In all CG-SDK systems, a time step of 20 fs was used, except for bilayers containing cholesterol, where a time step of 10 fs was used.

## MD systems setup

In order to study the formation of TG lenses (*Figure 1B*), bilayers containing different concentrations of TG were employed. To build these systems, boxes consisting of 50 PL molecules and a variable number of TG molecules, to see the effect of different TG concentrations on lens formation, were initially prepared. The two PL monolayers were displaced along the $z$-axis in order to allow the insertion of the TG molecules and avoid steric clashes due to bad contacts or overlapping molecules. The TG molecules were subsequently randomly placed between the two monolayers. Boxes were then replicated eight times along the $x$- and $y$-axes. Every system ultimately included a total of 3200 PL molecules. Different bilayer compositions were tested, as reported in *Table 1*. Simulations were run until spontaneous lens formation (see section *MD simulations analysis* for details) or for a total of 1.5

**Table 1.** List of all the molecular dynamics (MD) setups, with bilayer composition, number of triglyceride (TG) molecules, number of replicas, and length of simulations.

**TG lenses formation**

| System | Bilayer composition (no. of molecules) | No. of TG | No. of replicas | Length (μs) |
|---|---|---|---|---|
| 100% DOPC | 3200 DOPC | 64 | 3 | 1.5 |
| | 3200 DOPC | 128 | 3 | 1.5 |
| | 3200 DOPC | 192 | 3 | <1.5 |
| | 3200 DOPC | 256 | 3 | <1.5 |
| | 3200 DOPC | 320 | 3 | <1.5 |
| +60 mol% POPC | 1280 DOPC + 1920 POPC | 64 | 3 | 1.5 |
| | 1280 DOPC + 1920 POPC | 128 | 3 | 1.5 |
| | 1280 DOPC + 1920 POPC | 192 | 3 | <1.5 |
| | 1280 DOPC + 1920 POPC | 256 | 3 | <1.5 |
| | 1280 DOPC + 1920 POPC | 320 | 3 | <1.5 |
| + 60 mol% DPPC | 1280 DOPC + 1920 DPPC | 64 | 3 | 1.5 |
| | 1280 DOPC + 1920 DPPC | 128 | 3 | 1.5 |
| | 1280 DOPC + 1920 DPPC | 192 | 3 | <1.5 |
| | 1280 DOPC + 1920 DPPC | 256 | 3 | <1.5 |
| | 1280 DOPC + 1920 DPPC | 320 | 3 | <1.5 |
| + 60 mol% DLPC | 1280 DOPC + 1920 DLPC | 64 | 3 | 1.5 |
| | 1280 DOPC + 1920 DLPC | 128 | 3 | 1.5 |
| | 1280 DOPC + 1920 DLPC | 192 | 3 | 1.5 |
| | 1280 DOPC + 1920 DLPC | 256 | 3 | <1.5 |
| | 1280 DOPC + 1920 DLPC | 320 | 3 | <1.5 |
| + 60 mol% DOPE | 1280 DOPC + 1920 DOPE | 64 | 3 | 1.5 |
| | 1280 DOPC + 1920 DOPE | 128 | 3 | <1.5 |
| | 1280 DOPC + 1920 DOPE | 192 | 3 | <1.5 |
| | 1280 DOPC + 1920 DOPE | 256 | 3 | <1.5 |
| | 1280 DOPC + 1920 DOPE | 320 | 3 | <1.5 |
| +30 mol% DOPE | 2240 DOPC + 960 DOPE | 64 | 3 | 1.5 |
| | 2240 DOPC + 960 DOPE | 128 | 3 | 1.5 |
| | 2240 DOPC + 960 DOPE | 192 | 3 | <1.5 |
| | 2240 DOPC + 960 DOPE | 256 | 3 | <1.5 |
| | 2240 DOPC + 960 DOPE | 320 | 3 | <1.5 |
| + 60 mol% DLPE | 1280 DOPC + 1920 DLPE | 64 | 3 | 1.5 |
| | 1280 DOPC + 1920 DLPE | 128 | 3 | <1.5 |
| | 1280 DOPC + 1920 DLPE | 192 | 3 | <1.5 |
| | 1280 DOPC + 1920 DLPE | 256 | 3 | <1.5 |
| | 1280 DOPC + 1920 DLPE | 320 | 3 | <1.5 |
| +10 mol% Diacylglycerol (DAG) | 2880 DOPC + 320 DAG | 64 | 3 | 1.5 |
| | 2880 DOPC + 320 DAG | 128 | 3 | <1.5 |
| | 2880 DOPC + 320 DAG | 192 | 3 | <1.5 |
| | 2880 DOPC + 320 DAG | 256 | 3 | <1.5 |
| | 2880 DOPC + 320 DAG | 320 | 3 | <1.5 |
| +20 mol% DAG | 2560 DOPC + 640 DAG | 64 | 3 | 1.5 |
| | 2560 DOPC + 640 DAG | 128 | 3 | <1.5 |

*Table 1 continued on next page*

*Table 1 continued*

**TG lenses formation**

| System | Bilayer composition (no. of molecules) | No. of TG | No. of replicas | Length (μs) |
|---|---|---|---|---|
| | 2560 DOPC + 640 DAG | 192 | 3 | <1.5 |
| | 2560 DOPC + 640 DAG | 256 | 3 | <1.5 |
| | 2560 DOPC + 640 DAG | 320 | 3 | <1.5 |
| +10 mol% CHOL | 2880 DOPC + 320 CHOL | 64 | 3 | 1.5 |
| | 2880 DOPC + 320 CHOL | 128 | 3 | <1.5 |
| | 2880 DOPC + 320 CHOL | 192 | 3 | <1.5 |
| | 2880 DOPC + 320 CHOL | 256 | 3 | <1.5 |
| | 2880 DOPC + 320 CHOL | 320 | 3 | <1.5 |
| +20 mol% CHOL | 2560 DOPC + 640 CHOL | 64 | 3 | 1.5 |
| | 2560 DOPC + 640 CHOL | 128 | 3 | <1.5 |
| | 2560 DOPC + 640 CHOL | 192 | 3 | <1.5 |
| | 2560 DOPC + 640 CHOL | 256 | 3 | <1.5 |
| | 2560 DOPC + 640 CHOL | 320 | 3 | <1.5 |
| Endoplasmic reticulum (ER) like | 1856 DOPC + 960 DOPE + 192 CHOL + 192 DAG | 64 | 3 | 1.5 |
| | 1856 DOPC + 960 DOPE + 192 CHOL + 192 DAG | 128 | 3 | <1.5 |
| | 1856 DOPC + 960 DOPE + 192 CHOL + 192 DAG | 192 | 3 | <1.5 |
| | 1856 DOPC + 960 DOPE + 192 CHOL + 192 DAG | 256 | 3 | <1.5 |
| | 1856 DOPC + 960 DOPE + 192 CHOL + 192 DAG | 320 | 3 | <1.5 |
| Calculation of 'free TG' | | | | |
| 100% DOPC | 6050 DOPC | 1836 | 2 | 3 |
| + 60 mol% POPC | 2420 DOPC + 3630 POPC | 1836 | 2 | 3 |
| + 60 mol% DPPC | 2420 DOPC + 3630 DPPC | 1836 | 2 | 3 |
| + 60 mol% DLPC | 2420 DOPC + 3630 DLPC | 1836 | 2 | 3 |
| + 60 mol% DOPE | 2420 DOPC + 3630 DOPE | 1836 | 2 | 3 |
| + 30 mol% DOPE | 4235 DOPC + 1815 DOPE | 1836 | 2 | 3 |
| + 60 mol% DLPE | 2420 DOPC + 3630 DLPE | 1836 | 2 | 3 |
| +10 mol% DAG | 5445 DOPC + 605 DAG | 1836 | 2 | 3 |
| ER like | 3509 DOPC + 1815 DOPE + 363 DAG + 363 CHOL | 1836 | 2 | 3 |
| +20 mol% DAG | 4840 DOPC + 1210 DAG | 1836 | 2 | 3 |
| +10 mol% CHOL | 5445 DOPC + 605 CHOL | 1836 | 2 | 3 |
| +20 mol% CHOL | 4840 DOPC + 1210 CHOL | 1836 | 2 | 3 |
| Insertion of TG | 6050 DOPC | 1934 | 2 | 3 |
| Different sizes | 11,250 DOPC | 1836 | 2 | 3 |
| | 11,250 DOPC | 5508 | 2 | 3 |
| | 11,250 DOPC | 9180 | 2 | 3 |
| | 16,200 DOPC | 13,665 | 2 | 3 |
| Dissolution | | | | |
| Dissolution | 6050 DOPC | 50 | 1 | 0.8 |
| Effect of saturation in TG chains | | | | |
| DOPC+TOOP | 6050 DOPC | 1836 TOOP | 2 | 3 |
| DOPC+TOPP | 6050 DOPC | 1836 TOPP | 2 | 3 |
| Lateral pressure profile | | | | |
| DOPC | 3200 | 0 | 3 | 0.2 |

*Table 1 continued on next page*

*Table 1 continued*

**TG lenses formation**

| System | Bilayer composition (no. of molecules) | No. of TG | No. of replicas | Length (µs) |
|---|---|---|---|---|
| +10 mol% CHOL | 2880 DOPC + 320 CHOL | 0 | 3 | 0.2 |
| +20 mol% DAG | 2560 DOPC + 640 DAG | 0 | 3 | 0.2 |
| +10 mol% DAG | 2880 DOPC + 320 DAG | 0 | 3 | 0.2 |
| +30 mol% DOPE | 2240 DOPC + 960 DOPE | 0 | 3 | 0.2 |
| +20 mol% DOPE | 2560 DOPC + 640 DOPE | 0 | 3 | 0.2 |
| +60 mol% DLPC | 1280 DOPC + 1920 DLPC | 0 | 3 | 0.2 |
| +60 mol% DLPE | 1280 DOPC + 1920 DLPE | 0 | 3 | 0.2 |
| +60 mol% DPPC | 1280 DOPC + 1920 DPPC | 0 | 3 | 0.2 |

µs per run if no spontaneous lens formation was observed (*Table 1*). TG concentration is always reported as the ratio between TG and PL molecules.

The system with two regions with different DAG concentrations (*Figure 1E*) was obtained from a bilayer with a uniform mixture of 2880 DOPC and 320 DAG plus 64 TG molecules. After minimization, the DAG pool was divided into two different groups: one pool was free to move in all the regions of the bilayer, the other was subjected to a repulsive wall with a cylindric shape that prevented molecules to go back inside the cylinder once they crossed the wall in the opposite direction. The final systems contain two regions with different DAG concentrations coexisting in the same bilayer. The final concentration of DAG in the two regions is obtained as average concentration during production, after 100 ns of production (*Figure 1—figure supplement 1*).

To calculate the concentration of diluted TG in the bilayer (*Figure 2A,B, and G* and *Figure 3C,F, I,L,O*), systems were formed by positioning a lens of 1836 TG molecules between two monolayers with 3025 DOPC lipids each, as in *Ben M'barek et al., 2017*. We also created bigger systems, with 12,250 lipids and 1836, 5508, and 9180 TG molecules or with 16,200 DOPC and 13,665 TG, in order to study the size independence on the concentration of diluted TG in the bilayer at equilibrium (*Figure 2E*). All bilayer compositions described in *Table 1* were tested.

To study the effect of inserting new TG molecules in the bilayer (*Figure 2C*), we slightly separated both PL monolayers along the *z* direction to accommodate more TG, and then those TG molecules outside the previously formed lens (see section *MD simulations analysis*) were replicated three times using the TopoTools VMD plugin (*Kohlmeyer and Vermaas, 2017*).

To study the effect of unsaturation in TG chains on the equilibrium concentration of TG (*Figure 5A and B*), systems containing 6050 DOPC and 1836 TOOP (a TG molecule with two oleic acids and one palmitic acid) or 1836 TOPP (a TG molecule with one oleic acid and two palmitic acids) were built.

## MD simulations analysis

TG lens formation was defined when an aggregate of at least 25 TG molecules within a 3.5 nm distance cutoff of a TG molecule was stable for at least 5 ns. The simulations were run until the formation of a TG lens was observed or for 1.5 µs if no aggregation occurred. For each TG concentration, bilayer composition, and setup, three independent simulations were performed. The rate of formation was calculated, from the average over the three replicas, as the inverse of the time of formation. Error bars were computed using error propagation as the standard deviation of the time of formation from the three independent simulations. For simulations where no formation was observed, the error bars were given as the inverse of the total simulation time (1.5 µs).

The analysis of the concentration of diluted TG was performed using the protocol outlined below. We defined a lens as a set composed by all the TG molecules found within a 5 nm distance of another TG molecule and not within 2.8 nm of a PL. The 'lens-free bilayer' was accordingly defined as all lipid molecules that were at least 2.5 nm away from the lens. We tested different selections and we chose the selection at which the diluted TG concentration was converged (*Figure 2—figure*

*supplement 1*) and that allowed us to consider the widest area of lens-free bilayer, excluding the side of the lens where TG molecules moved continuously inside and outside the lens (*Figure 2—figure supplement 1*). The simulations were run for 3 μs (see *Table 1*). The percentage of TG molecules inside the bilayer was obtained averaging the values of the last 1.5 μs of simulation, and the corresponding error bars were obtained from the standard deviation over two independent simulations. TG concentration is always reported as the ratio between TG and PL molecules. The dimensions of TG lenses were calculated using the minimum and the maximum *x*, *y*, and *z* coordinates of the TG molecules at each side of the lens, averaged over time.

Density maps (*Figure 4A–C*) were calculated using GROMACS tools (*Van Der Spoel et al., 2005*). To obtain an enrichment/depletion map for DOPE, the density map of DOPE was divided by the sum of the maps of DOPE and DOPC and then divided by the concentration of DOPE in bulk. The same approach was used to obtain enrichment/depletion map for DAG and DPPC.

The relative energy of nucleation ($E_{nucl}$) was obtained from the rate of formation of blisters in bilayers containing 6% TG (*Figure 4D*) with

$$\text{nucleation rate} \propto e^{-E_{nucl}/KT}$$

The values are reported as relative $E_{nucl}$, considering DOPC bilayers as reference ($E_{nucl}$ [(composition)]/$E_{nucl}$ [(DOPC)]). The error bars are relative errors.

For the calculation of LPPs (*Figure 4E*), bilayers not containing TG molecules were equilibrated for a variable length (from 200 to 1500 ns) and then run for additional 200 ns for the calculation of LPP. Stress components along *z* were obtained using the LAMMPS commands 'compute stress/atom' and 'fix ave/chunch' during production and then converted to corresponding pressures using the script stress2press.py, available in the ELBA-LAMMPS toolkit (https://github.com/orsim/elba-lammps/tree/master/tools). The LPP curves are reported as average and the error bars as standard deviation over three independent replicas. $\pi_{CH}$ values were obtained integrating LPP curves for $\pi(z)$ values > 0, employing the Simpson's rule and using a bin size of 0.01 Angstroms. The values of $\pi_{CH}$ and the relative percentage change with DOPC bilayers as reference are reported in *Supplementary file 1*.

Spontaneous curvatures, $c_0$, for the bilayers were computed from the first moment of the LPPs using the following *Safran, 2003*:

$$-\int_0^{L_z/2} z[P_T(z) - P_N]dz = \kappa_b c_0$$

where $P_N$ and $P_T$ are the normal and tangential components of the pressure tensor with respect to the bilayer surface, the limits of the integral correspond to one of the membrane leaflets (assuming that the bilayer midplane is located at $z = 0$ and its thickness is equal to $L_z$), and $\kappa_b$ is the monolayer bending moduli.

The required monolayer bending moduli were estimated using two different approaches: (i) from the real-space analysis of the instantaneous surface deformations and (ii) from the Fourier-space analysis of the membrane fluctuations (*Allolio et al., 2018*). Notably, both methodologies led to similar (and reasonable, $\kappa_b$ being of the order of 10 $k_B$T) values (see *Supplementary file 1*), which somehow serves as further validation of the reliability of the obtained results. The values of $\kappa_b$ used for the calculations of the monolayer curvature stress (*Figure 4G*) are from the Fourier-space analysis of the membrane fluctuations.

## Yeast strains and growth conditions

Triple-mutant strain *pah1Δ are1Δ are2Δ* was generated by gene disruption, using PCR deletion cassettes and a marker rescue strategy (*Longtine et al., 1998*). Quadruple-mutant strain, are1Δ are2Δ dga1Δ lro1Δ, was generated by mating and sporulation. Genotype of yeast strains is shown in *Supplementary file 1*. Strains were cultivated in YP-rich medium (1% bacto yeast extract, 2% bacto peptone [USBiological Swampscott, MA]) or selective (SC) medium (0.67% yeast nitrogen base without amino acids (USBiological), 0.73 g/L amino acids) containing 2% glucose. Fatty acid-supplemented media contained either lauric, palmitic, or palmitoleic acid (Sigma-Aldrich, St Louis, MO) and 1% Brij 58.

## Fluorescence microscopy

BODIPY fluorescence was recorded using a Visitron Visiscope CSU-W1 (Visitron Systems, Puchheim Germany), consisting of a CSU-W1 spinning disk head with a 50-µm pinhole disk (Yokogawa, Tokyo, Japan) and Evolve 512 (Photometrics) EM-CCD camera mounted to a Nikon Ti-E inverted microscope equipped with a PLAN APO 100× NA 1.3 oil objective (Nikon) and controlled by Visitron Visi-View software 4.2.0. Cells were grown in SC media and stained with BODIPY 493/503 (1 µg/mL) for 5 min at room temperature. Images were recorded using identical microscope settings. BODIPY fluorescence was measured using ImageJ and membrane-associated fluorescence was calculated by subtracting LD-associated fluorescence from whole cell fluorescence. All experiments were repeated three times.

## Image rendering

All MD images were rendered using VMD software (*Humphrey et al., 1996*) and graphs were generated using the software MATLAB.

Experimental images were treated using Image-J software and then resized in Adobe Photoshop CC 2015. Microscopic experiments were performed three times with similar results.

All source data, input files for MD simulations, and statistical analyses for *Figure 5* can be found at: http://doi.org/10.5281/zenodo.4457468.

## Acknowledgements

We thank Vikram Reddy Ardham and Vineet Choudhary for useful discussions. This work was supported by the Swiss National Science Foundation (grant #163966). This work was supported by grants from the Swiss National Supercomputing Centre (CSCS) under project ID s726, s842 and s980. We acknowledge PRACE for awarding us access to Piz Daint, ETH Zurich/CSCS, Switzerland. RS was supported by the Swiss National Science Foundation (31003A_173003) and the Novartis Foundation for medical-biological Research (19B140).

## Additional information

### Funding

| Funder | Grant reference number | Author |
| --- | --- | --- |
| Schweizerischer Nationalfonds zur Förderung der Wissenschaftlichen Forschung | 163966 | Valeria Zoni<br>Pablo Campomanes<br>Stefano Vanni |
| Novartis Stiftung für Medizinisch-Biologische Forschung | 19B140 | Roger Schneiter |
| Schweizerischer Nationalfonds zur Förderung der Wissenschaftlichen Forschung | 31003A_173003 | Roger Schneiter |

The funders had no role in study design, data collection and interpretation, or the decision to submit the work for publication.

### Author contributions

Valeria Zoni, Software, Investigation, Writing - review and editing; Rasha Khaddaj, Investigation; Pablo Campomanes, Software, Investigation, Methodology; Abdou Rachid Thiam, Conceptualization, Writing - review and editing; Roger Schneiter, Conceptualization, Supervision, Funding acquisition, Project administration, Writing - review and editing; Stefano Vanni, Conceptualization, Resources, Supervision, Funding acquisition, Investigation, Methodology, Writing - original draft, Project administration, Writing - review and editing

### Author ORCIDs

Valeria Zoni  https://orcid.org/0000-0002-5370-0356
Rasha Khaddaj  https://orcid.org/0000-0003-2951-3195

Pablo Campomanes (iD) https://orcid.org/0000-0001-9229-8323
Abdou Rachid Thiam (iD) https://orcid.org/0000-0001-7488-4724
Roger Schneiter (iD) https://orcid.org/0000-0002-9102-8396
Stefano Vanni (iD) https://orcid.org/0000-0003-2146-1140

### Decision letter and Author response
Decision letter https://doi.org/10.7554/eLife.62886.sa1
Author response https://doi.org/10.7554/eLife.62886.sa2

## Additional files

### Supplementary files
• Supplementary file 1. Additional information. Supplementary table 1. Non-bonded parameters derived for this study. Supplementary table 2. Calculated values of $\pi_{CH}$ for each bilayer composition and the relative percentage change with respect to the composition '100% DOPC'. Supplementary table 3. Calculated values of $\kappa_b$ for each bilayer composition with two different methods: from the real-space analysis of the instantaneous surface deformations ($\kappa_b$ ReSIS) and from the Fourier-space analysis of the membrane fluctuations ($\kappa_b$ Fluct). Supplementary table 4: *S. cerevisiae* strains used in this study.

• Transparent reporting form

### Data availability
Data Availability: All source data, input files for MD simulations and statistical analyses can be found at the following DOI: http://doi.org/10.5281/zenodo.4457468.

The following dataset was generated:

| Author(s) | Year | Dataset title | Dataset URL | Database and Identifier |
|---|---|---|---|---|
| Zoni V, Khaddaj R, Campomanes P, Thiam AR, Schneiter R, Vanni S | 2021 | Pre-existing bilayer stresses modulate triglyceride accumulation in the ER versus lipid droplets | http://doi.org/10.5281/zenodo.4457468 | Zenodo, 10.5281/zenodo.4457468 |

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
