## [Decision Letter]

**Acceptance summary:**

The mechanism for the generation of lipid droplet (LD) from ER membrane remains debated. This article presents the results of MD coarse-grained simulations of phospholipid bilayers containing different amounts of triglyceride (TG) molecules. A few fluorescence experiments on yeast support the predictions of the model. It proposes that the initial generation of lenses/blisters, precursors of LDs, results from pre-existing lateral stresses in the hydrophobic core of the ER membrane that are relaxed by the condensation of TG in lenses. This effect is modulated by the nature of the acyl chains of the phospholipids present in the membrane.

**Decision letter after peer review:**

[Editors’ note: the authors submitted for reconsideration following the decision after peer review. What follows is the decision letter after the first round of review.]

Thank you for submitting your work entitled "Lipid droplet biogenesis is driven by liquid-liquid phase separation" for consideration by *eLife*. Your article has been reviewed by three peer reviewers, and the evaluation has been overseen by a Reviewing Editor and a Senior Editor. The following individual involved in review of your submission has agreed to reveal their identity: Joel Goodman (Reviewer #2).

Our decision has been reached after consultation between the reviewers. Based on these discussions and the individual reviews below, we regret to inform you that your work will not be considered further for publication in *eLife*.

Identification of the mechanism for lipid droplet (LD) generation is an important and popular question. The proposition that LD nucleation and formation can be considered as a liquid-liquid phase separation is a hot notion in current Cell Biology. The authors used mostly simulations to address these points; reviewers salute their expertise and recognize that this approach could be useful for making novel predictions about factors that may affect droplets. However, reviewers considered that there is not enough new information in the manuscript to warrant publication in *eLife* and a physical mechanism is still needed. Moreover, they think that the use of the term LLPS can be misleading to the general biology audience. Eventually, there were also technical issues about the model and the spontaneous curvature of some lipids.

[Editors’ note: further revisions were suggested prior to acceptance, as described below.]

Thank you for submitting your article "Hydrophobic bilayer pressure modulates triglyceride accumulation in the ER versus lipid droplets" for consideration by *eLife*. Your article has been reviewed by three peer reviewers, and the evaluation has been overseen by Patricia Bassereau as Reviewing Editor and José Faraldo-Gómez as Senior Editor. The following individuals involved in review of your submission have agreed to reveal their identity: Joel Goodman (Reviewer #1); Peter Olmsted (Reviewer #3).

The reviewers have discussed the reviews with one another and the Reviewing Editor has drafted this decision to help you prepare a revised submission.

As the editors have judged that your manuscript is of interest, but also that additional calculations are required before it can be published, we would like to draw your attention to changes in our revision policy that we have made in response to COVID-19 (https://elifesciences.org/articles/57162). First, because many researchers have temporarily lost access to the labs, we will give authors as much time as they need to submit revised manuscripts. We are also offering, if you choose, to post the manuscript to bioRxiv (if it is not already there) along with this decision letter and a formal designation that the manuscript is "in revision at *eLife*". Please let us know if you would like to pursue this option. (If your work is more suitable for medRxiv, you will need to post the preprint yourself, as the mechanisms for us to do so are still in development.)

Summary:

The mechanism for the generation of lipid droplet (LD) from ER membrane remains debated. This article presents the results of coarse-grained MD simulations of phospholipid bilayers containing different amounts of triglyceride (TG) molecules. A few fluorescence experiments on yeast support the predictions of the model. It proposes that the initial generation of lenses/blisters, precursors of LDs, results from pre-existing lateral stresses in the hydrophobic core of the ER membrane that are relaxed by the condensation of TG in lenses. This effect is modulated by the nature of the acyl chains of the phospholipids present in the membrane.

Essential revisions:

The reviewers have appreciated the novel version of your manuscript with a reduced emphasis on liquid-liquid phase separation and rather on molecular mechanisms that account for the TG accumulation in blisters. However, they consider that your work could be reinforced by a more quantitative analysis of your mechanism based on the pressure profile and by revising the discussion of the effects of TG on phase separation to explain that the effects of TG are both to change the chemical potential in the flat phase, as well as to potentially change the mechanical energy of the lens with which it coexists. They have also stressed that thermodynamics and kinetics arguments are too much entangled in the current version of the manuscript. More precisely, they recommend that you perform the following calculations:

1) Curvature stress can be determined from the pressure profile. The theory is available to determine if the change in tension (or chemical potential) is sufficient to change the observed rates under different lipid conditions. It should not be problematic since it is a trivial numerical integral of the first moment of the lateral pressure profile (assuming a reasonable bending modulus).

2) You should determine, at least qualitatively, the relative effect of chemical potential and curvature on kinetics, using available theory.

[Editors' note: further revisions were suggested prior to acceptance, as described below.]

Thank you for submitting your article "Pre-existing bilayer stresses modulate triglyceride accumulation in the ER *versus* lipid droplets" for consideration by *eLife*. Your article has been reviewed by three peer reviewers, and the evaluation has been overseen by Patricia Bassereau as Reviewing Editor and José Faraldo-Gómez as Senior Editor. The following individuals involved in review of your submission have agreed to reveal their identity: Joel Goodman (Reviewer #1); and Peter Olmsted (Reviewer #3).

Based on the reviewers' comments and a subsequent consultation among editors and reviewers, and given the changes in the *eLife* revision policy in response to the COVID-19 pandemic (https://elifesciences.org/articles/57162), we would like to invite you to submit a revised version of your article that addresses issues of clarity and presentation – as noted below – without the need for additional data.

Summary:

Cytoplasmic lipid droplets emerge from the endoplasmic reticulum. This paper combines molecular dynamics simulations with experimental studies on yeast to show that phase separation plays a role in the process. The authors demonstrate their claim by changing the fraction of triglycerides, and by studying the curvature and curvature stresses in the lipid droplets as a function of lipid composition. They show how different features of the lipids, such as lipid tail length and degree of saturation, can modulate the formation and the stability of lipid droplets in the endoplasmic reticulum.

Revisions:

Reviewers and editors agree that you have satisfactorily integrated the main changes that were recommended at the previous stage. However, prior to acceptance for publication, we think that you should acknowledge the ambiguity in the relationship between curvature stress and chemical potential, and soften the conclusion that the chemical potential is a complete descriptor.

---

## [Author Response]

[Editors’ note: the authors resubmitted a revised version of the paper for consideration. What follows is the authors’ main response to the first round of review.]

We thank the reviewers for substantially helping us improve the manuscript. Concerning their main criticisms, the new version of the manuscript:

Proposes an entirely new physical mechanism to understand and predict why certain lipid composition would promote triglyceride (TG) storage in LD *vs* their accumulation in the ER. This mechanism is described in the new Figure 4. In detail we show that the pre-existing molecular interactions between lipids’ acyl chains, and namely their hydrophobic pressure, drive this process. This property can be quantified in MD simulations from the lateral pressure profiles of TG-free lipid bilayers, and this quantity correlates very well with the observed propensity of TG to accumulate in the bilayer.Does not contain any reference to LLPS, as this could indeed be misleading.Provides additional experimental validation of our model (Figure 6) and a new and improved quantification of the experimental observations.Clarifies potential technical issues.

[Editors’ note: what follows is the authors’ response to the second round of review.]

Essential revisions:The reviewers have appreciated the novel version of your manuscript with a reduced emphasis on liquid-liquid phase separation and rather on molecular mechanisms that account for the TG accumulation in blisters. However, they consider that your work could be reinforced by a more quantitative analysis of your mechanism based on the pressure profile and by revising the discussion of the effects of TG on phase separation to explain that the effects of TG are both to change the chemical potential in the flat phase, as well as to potentially change the mechanical energy of the lens with which it coexists. They have also stressed that thermodynamics and kinetics arguments are too much entangled in the current version of the manuscript. More precisely, they recommend that you perform the following calculations:1) Curvature stress can be determined from the pressure profile. The theory is available to determine if the change in tension (or chemical potential) is sufficient to change the observed rates under different lipid conditions. It should not be problematic since it is a trivial numerical integral of the first moment of the lateral pressure profile (assuming a reasonable bending modulus).2) You should determine, at least qualitatively, the relative effect of chemical potential and curvature on kinetics, using available theory.

To address these comments, we have followed the reviewers’ suggestions and we have:

1) Extracted the spontaneous curvature c_0_ from the first moment of the lateral pressure profiles by using two different approaches to estimate the corresponding bending moduli κ_B_ (from a real-space analysis of the instantaneous surface deformations and from the Fourier space analysis of the membrane fluctuations (see Allolio et al., 2018). Both methodologies led to similar (and reasonable, with monolayer bending rigidities around 10 K_B_T) results, which somehow serves to validate the reliability of the obtained values. This allowed us to estimate curvature stress for the various lipid bilayers under investigation and to determine whether pre-existing curvature stresses could explain the differences in chemical potential. Indeed, we found a good correlation between curvature stress and TG chemical potential. This is not totally surprising as curvature stress is related, through the lateral pressure profile, to hydrophobic tension, the property we previously identified as proportional to TG chemical potential.

We have thus opted to discuss more generally changes in chemical potential as a result of bilayer stresses (curvature, hydrophobic tension), including in the title, and we have further stated that changes in chemical potential seem proportional to changes in internal tension consistently with a simple Gibbs-Duhem relationship, as the reviewers correctly pointed out.

2) We have further contextualized the relative effect of curvature and chemical potential on kinetics by discussing available theory and contextualizing our results in this context. Notably, we found that changes in chemical potential appear sufficient to describe the observed rates in different lipid conditions, at least at the TG concentrations investigated in this work (Figure 4D). Rather, curvature stress indirectly affects blister formation kinetics via its effect on TG chemical potential, as monolayer curvature stress strongly correlates with TG chemical potential (Figure 4G). Interestingly, this is consistent with recent experimental results (Santinho et al., 2020) showing that the effect of membrane curvature in LD formation originates from its ability to alter the critical TG concentration required for the spontaneous condensation of TGs. Finally, we also further stressed that nucleation rates, in vivo, are likely to be mostly determined by protein activity (Prasanna et al., BioRxiv, 2020; Zoni et al., BioRxiv 2020). To clarify these aspects, we have added this entire part:

“On the other hand, the nucleation energy for nascent TG blisters can be described as a sum of two contributions, E_nucl_ = E_∆μ_+E_s_, with E_∆μ_ arising from TG de-mixing and E_s_ arising from interfacial and mechanical contributions. […] Notably, this is consistent with recent experimental results showing that the effect of membrane curvature in LD formation originates from its ability to alter the critical TG concentration required for the spontaneous condensation of TGs”

And later

“To better quantify the relationship between TG concentration and lipids’ acyl chains properties, we plotted the equilibrium TG concentration computed in our simulations against two distinct properties: (i) the hydrophobic pressure π_CH_ , *i.e.* the integral of the LPP corresponding to the area of the positive region (π(z) > 0) between both glycerol minima in the LPPs (Figure 4F)(*40*) and (ii) the total monolayer curvature stress κ_b_ c_0_^2^ , where κ_b_ is the monolayer bending rigidity and c_0_ its spontaneous curvature c_0_ (Figure 4G). In both cases, we found a very good correlation, suggesting that pre-existing stresses in the bilayer can predict the propensity of the bilayer to accept increasing concentration of TG molecules, and hence oppose or promote their spontaneous de-mixing to form oil blisters.”

Furthermore, throughout the text we have further clarified thermodynamics and kinetics arguments and considerations. For example, here:

“Our *in silico* reconstitution of TG blister formation suggests that this mechanism is consistent with a nucleation process (Figure 1B-C). […] To this end, we sought to determine the equilibrium concentration of TG molecules in the bilayer (*i.e.* their chemical potential) when the bilayer is in equilibrium with a TG droplet large enough to mimic the properties of a bona-fide condensed phase.”

Or here:

“The observation that DAG concomitantly promotes TG blister formation and depletion of TG molecules from the ER is intriguing. To further characterize the relationship between blister formation and TG chemical potential, we next focused on the role of membrane lipids on this process, with the aim of potentially establishing a correspondence with the role of ER membrane in modulating LD formation”.

We thank the reviewer for pointing out this potential confusion as it is particularly important for us to be as clear as possible on this point. In fact, our current working model is that LD nucleation is largely driven by proteins, as recent preprints seem to also suggest (Prasanna et al., BioRxiv, 2020; Zoni et al., BioRxiv 2020). On the contrary, this work shows that the extent of TG accumulation in the ER is driven by lipids. Hence, we believe that the information on thermodynamics aspects that we can extract from the MD simulations are (i) quantitatively more accurate and (ii) more physiologically-relevant than the information we can extract on kinetic processes.

[Editors' note: further revisions were suggested prior to acceptance, as described below.]

Revisions:Reviewers and editors agree that you have satisfactorily integrated the main changes that were recommended at the previous stage. However, prior to acceptance for publication, we think that you should acknowledge the ambiguity in the relationship between curvature stress and chemical potential, and soften the conclusion that the chemical potential is a complete descriptor.

We thank the Editors and the reviewers for appreciating our work, and for giving us the possibility to further clarify some remaining ambiguity in the relationship between curvature stress and TG chemical potential with regard to blister nucleation.

We observe that TG chemical potential and curvature stress are highly correlated (Figure 4G). This is not surprising as in current theory of nucleation phenomena in the context of LDs (see Table 1 in Thiam and Foret, *BBA,* 2016), several factors affect both the contribution of chemical potentials and membrane mechanics, which are therefore not uncoupled. Accordingly, our simulations show that the two terms are, indeed, not independent. Rather, as TG synthesis occurs in the ER upon specific cellular stimuli, we can conclude that, from a temporal perspective, bilayer properties precedes the presence of TG molecules, and hence we can postulate a temporal causality, implying that curvature stress affects TG chemical potential. This is why our main conclusion is that bilayer stresses (including curvature stress) are responsible for modulating LD formation. This is also evident from the title of the manuscript.

Because of the interlink between TG chemical potential and curvature stress, trying to estimate the independent contribution of each towards blister nucleation kinetics would require a new effective theory where curvature stress not only contributes to nucleation energy directly, but also indirectly through its effect on TG chemical potential. While of clear interest to the field, we think that this is not only beyond the scope of this work, but that it would also require the accurate estimation of additional terms (e.g rim energy, monolayer tension [Deslandes et al., 2017], bilayer tension [M’Barek et al., 2017]) where MD simulations have intrinsic limitations because of the use of periodic boundary conditions and fixed bilayer tension.

Rather, as bilayer stresses modulate TG chemical potential, we find that this property is a good “effective” descriptor of blister nucleation and, more importantly, of the equilibrium coexistence between TG blisters and diluted-TG in the bilayer. Of note, this is very important as TG concentration is easier to measure both *in silico* (as it is an equilibrium property) as well as in vitro (Santinho et al., 2020) and in vivo (Figures 5 and 6).

Hence, in this new version of the manuscript we have now further and better clarified the relationships between curvature stress and TG chemical potential in various parts of text (highlighted in red). For example, here:

“On the other hand, the nucleation energy for nascent TG blisters can be described as a sum of two linked contributions, E_nucl_ = E_Dµ_+E_s_, with E_Dµ_ arising from TG de-mixing and E_s_ arising from interfacial and mechanical contributions. Hence, the effect of PL on phase separation can be both on TG chemical potential and membrane mechanics: PL may induce curvature stress in the bilayer that could potentially alter TG chemical potential, and generate interactions with TG that could alter both monolayer and bilayer bending rigidity.”

or here:

“In addition, the observation that TG chemical potential and monolayer curvature stress are correlated (Figure 4G) confirms that mechanical and de-mixing energetic contributions are not independent, as TG de-mixing is modulated by bilayer mechanical properties.”

As requested, we also softened our conclusion that chemical potential is a “complete” descriptor, rather stressing that it is a “good” descriptor that can be easily estimated by means of different computational and experimental techniques, for example here:

“TG chemical potential is a good descriptor of blister formation kinetics.” (as opposed to “is sufficient to describe”)

Or here:

“The nucleation energy decreased with excess TG in the bilayer, which agrees with the description of nucleation phenomena. Therefore, this observation supports that the simple increase in TG bilayer chemical potential, here, through increasing TG bilayer concentration, is sufficient to stimulate LD nucleation. Hence, the observed changes in nucleation rate as a function of lipid composition pertain, at least partly, to a variation in TG bilayer chemical potential (Figure 4D).”

Or by removing the following sentence:

“These results indicate that energetic terms arising from membrane remodeling and interfacial interactions appear to be secondary to TG chemical potential towards interpreting kinetic aspects of TG blister formation in silico”